# Education from the crib on: The potential of the Newborn Cohort of the German National Educational Panel Study

Journal of
open psychology data

DATA PAPER

]u[ ubiquity press

**MANJA ATTIG** (iD)

**MARKUS VOGELBACHER**

**SABINE WEINERT** (iD)

*Author affiliations can be found in the back matter of this article

## ABSTRACT

The Newborn Cohort of the German National Educational Panel Study provides longitudinal data for about 3,500 newborns and their families. Starting in 2012, nine annual waves have been published until 2022.

The data sets include information on domain-specific and general competencies from standardized tasks/tests with children, observational data on semi-standardized parent-child-interaction, as well as data from parent's and educational institution's interviews/questionnaires.

The data is accessible via the Research Data Center of the LIfBi and comprehensively documented (English, German) to be used, e.g., in research on child development, educational trajectories, as well as on facets of longitudinally assessed learning environments.

**CORRESPONDING AUTHOR:**
**Manja Attig**

Leibniz Institute for
Educational Trajectories, DE

manja.attig@lifbi.de

**KEYWORDS:**
Education; Newborn Cohort;
Competence Development;
Longitudinal

**TO CITE THIS ARTICLE:**
Attig, M., Vogelbacher, M., &
Weinert, S. (2023). Education
from the crib on: The potential
of the Newborn Cohort of the
German National Educational
Panel Study. *Journal of Open
Psychology Data,* 11: 13,
pp. 1–18. DOI: https://doi.
org/10.5334/jopd.81

## (1) BACKGROUND

In the first years of life the foundations for later educational processes and trajectories are laid (Belsky et al., 2007). When entering school, children already differ in their language, (pre-) reading, and math competence (NICHD ECCRN, 2005; Weinert, Ebert, & Dubowy, 2010). Skills – such as language, early literacy, and numeracy – are strong predictors of later academic success (e.g., Duncan et al., 2007; Melhuish, Phan, Sylva, Siraj-Blachford, & Taggart, 2008). Hence, education starts not with school entry but already from the very first years onward. To take this into account, the German National Educational Panel Study (NEPS; Blossfeld & Rossbach, 2019) includes a Newborn Cohort Study (Starting Cohort 1; SC1; e.g., NEPS Network, 2022; Weinert, Linberg, Attig, Freund, & Linberg, 2016) to investigate educational trajectories and the development of competencies from early on. The research potential of these data will be introduced in the present paper.

Theoretical models such as the bioecological model of development (Bronfenbrenner & Morris, 2006) emphasize the interplay of multiple contexts in shaping children's development. Hence not only children's own abilities and their prerequisites play a role for their development but also the learning environments they grow up in and interact with. In line with these theoretical frameworks, research has shown that children's cognitive development and educational career are influenced by characteristics of the family and home learning environment (Anders et al., 2012; Melhuish et al., 2008; Sirin, 2005; Weinert & Ebert, 2013). In agreement with international results (e.g., Hart & Risley, 1995), analyses with the data of the German BiKS-3-18 study[1] documented significant social disparities in children's early language skills and their domain-specific knowledge at the age of 3 years (Dubowy, Ebert, von Maurice, & Weinert, 2008; Kurz, von Maurice, Dubowy, Ebert, & Weinert, 2008). Furthermore, significant effects on child development are not only documented for the home learning environment (Attig & Weinert, 2020; Lehrl, Ebert, Blaurock, Rossbach, & Weinert, 2020) but also for preschool quality (e.g., Anders, Große, Rossbach, Ebert, & Weinert, 2013; Sylva, Melhuish, Sammons, Siraj-Blatchford, & Taggart, 2011).

Summing up, research has shown that individual differences arise early and were – amongst others – influenced by learning environments (Hart & Risley, 1995). However it's still necessary to learn more about the development of children, their learning environments, their interrelation with others, and most importantly the underlying mechanisms of how – for example – the social background impacts child development. For this, longitudinal studies starting in the early years are needed to empirically address these kinds of questions. To the knowledge of the authors, in Germany only a few longitudinal studies exist which investigate educational development already early in life (e.g., German BiKS-3-18 study; starting with 3-year-old children; Weinert, Rossbach, Blossfeld, & Artelt, in press). With the SC1, the NEPS takes up this challenge and accompanied infants from their first year of life.

The NEPS is a nationally representative educational study established and carried out by a nation-wide interdisciplinary scientific network of researchers. The goal is to collect high-quality longitudinal data to investigate, monitor, and compare competence development and educational processes in Germany across the whole lifespan from early childhood until late adulthood. To achieve this goal, a longitudinal multi-cohort sequence design was developed to collect data simultaneously from currently six different starting cohorts (Newborn Cohort, Kindergarten Cohort, Cohorts starting in Grade 5, in Grade 9; with first-year university students, and adults). In addition to educational stage-specific topics (e.g., the use of early childcare), the following six general topics (pillars of the NEPS) are addressed in all cohorts of the NEPS and, in the following, exemplified with respect to the SC1:

○ Competence development: domain-specific and domain-general indicators of early child capacities, characteristics, and developments (e.g., mathematical competence, vocabulary, executive functions)
○ Personality and motivation (e.g., temperament, socio-emotional development)
○ Learning environments (e.g., structural and process characteristics of different learning environments)
○ Factors impacting educational decisions (e.g., educational aspirations, social and cultural capital)
○ Migration (e.g., language of origin, country of origin, integration)
○ Monetary and nonmonetary returns to education (e.g., health, well-being and satisfaction of the parents, income into society).

Special attention is given to the assessment of social background factors (e.g., parent's education, occupational prestige, and family income).

The SC1 is the youngest cohort of the NEPS and started in 2012 with around 3,500 representatively drawn 7 months old infants and their families (Weinert et al., 2016; Würbach, Zinn, & Aßmann, 2016). At least once a year, an assessment with the children and their families took place, mainly in the households of the families. With already 9 assessment waves published on the children and their families, the SC1 offers a rich dataset to address research questions from a developmental and educational psychological perspective, e.g., in-depth analyses of developmental progress and the emergence of social disparities in children's (early) competence development. As an example, Attig and

Weinert (2020) investigated the effects of the home learning environment in the first two years and their impact on early language skills at the age of two. The authors used codings of the parent-child interaction (conducted at wave 1 to 3) as well as the frequency of joint picture book reading as process characteristics of the home learning environment. As a second example, Huang and colleagues (2022) investigated protective and risk pathways on children's behavioral difficulties drawing on two national samples from the United Kingdom (Millenium Cohort) and Germany (NEPS). From the NEPS, the authors used data of the parent-child interaction, the tested vocabulary of the children as well as information from the parent interview (e.g. behavioral difficulties). These are just two examples of the broad range of relevant constructs in the SC1 for psychologically oriented research questions.

## (2) METHODS

Most assessment of the SC1 of the NEPS took place in the households of the families. All data collections in the families were carried out by a survey institute in close collaboration with the NEPS-network. To keep the burden of the families low, each assessment in the household should not exceed 90 minutes. In wave 1, the mothers were intended to be the primary respondents as only they could provide valid information about their conditions and feelings during and after pregnancy. As changes of the respondent person in the following waves were handled strictly, only a few changes of the primary respondents occurred.

### 2.1 STUDY DESIGN

As all cohort studies of the NEPS, the SC1 is a longitudinal study. It started with the first assessment wave in 2012 when the infants were 6 to 8 months old. Different modes were used to collect the data in SC1, namely direct assessments of child development via competence tests and standardized and semi-standardized observations of early prerequisites/capacities (e.g., early child attention, sensorimotor development, interactional behavior),[2] computer-assisted personal interviews (CAPI), computer-assisted telephone interviews (CATI) and paper-and-pencil questionnaires (PAPIs) for one parent of each child as well as PAPIs for educational professionals.

In addition to the data collection with the parent, semi-standardized videotaped observational measures (e.g., parent-child interaction) were administered in the first three waves (see 2.5. for a more detailed description of the situation). From wave 4 on, also more functional competence tests (e.g., longitudinal coherent assessment of vocabulary knowledge, mathematical and science competence) as well as educational stage-specific tests or standardized tasks (e.g., delay of

gratification, executive functions and memory) with the child were administered using a tablet computer (see Hachul et al., 2019; Weinert et al., 2019 for overviews; for an overview of the time points of all measurements see Table 2).

Table 1 gives an overview of all waves, the used modes and the respective child age. In the first two years, three waves took place. Beginning with age 3 (wave 4), a survey was administered every year. In nearly every wave, the assessments were carried out in the households of the families including a CAPI with the parent and direct assessments with the child. The only exception was wave 2. In wave 2, all families took part in a CATI and due to design decisions only half of the sample was surveyed additionally in the children's home (observational measures with the child).[3] Beginning with wave 3, all families who could not be contacted for the CAPI household survey or preferred a telephone interview instead of the personal interview in their households were converted to a CATI-survey (around 5% of the interviews took place as telephone interview across the waves). In this case, the competence tests and observational measures with the child were not conducted. Between wave 7 and 8, most of the children of the SC1 changed from preschool to primary school. In spring 2019, 96% of the children in SC1 attended primary school (assessment wave 8). The assessments in wave 9, initially planned as regular computer-assisted parent interview with tablet-based competence measures of the children in the families' households, was stopped shortly after beginning of the field due to the first lockdown of the COVID-19 pandemic in March 2020. The data collection restarted in June 2020 as telephone interview carried out by the already trained regular CAPI-interviewers for the parents (CATI-Remote). The restarted CATI-Remote survey included a special module which covered questions regarding the COVID-19 pandemic and the families dealing with the school shutdown. The mode of the competence tests with the children was changed to online tests with simultaneous support by interviewers by telephone (CAPI by Phone). Please note that the data collection for the cohort is not finished yet, the children and their families are still followed up in further waves.

As additional context person, childminders (in wave 2 and 3), educators (in wave 2 until wave 7), and institutional manager (in wave 4 until wave 7) were sent a questionnaire. Due to data protection issues the questionnaires couldn't be sent directly to the childminders or child care institutions. Hence, the families were ask to give the questionnaires to the educators or the childminder of their child. Then, the child care institutions were asked to send the questionnaire back to the survey institute. The response rate varied between 21% and 41% with an average of 29% for educators, 24% for institutional manager and 36% for childminders.

| WAVE | INTERVIEW-MODE | DIRECT ASSESSMENTS OF CHILD AND HOME LEARNING ENVIRONMENT (HLE) | YEAR | CHILD (TARGET) AGE | PARTICI-PATING FAMILIES | CHILDCARE | | | SUF RELEASE |
|------|------|------|------|------|------|------|------|------|------|
| | | | | | | EDU-CATOR | INSTITU-TION MANAGER | CHILD-MINDER | |
| 1 | CAPI | Standardized and semi-standardized observational measures (incl. HLE) | Aug-2012–Mar 2013 | 7 months | 3,481 | | | | 2015 |
| 2 | CATI, PAPI | | Apr–Oct 2013 | 14 months | 2,862 | 171 | | 73 | 2015 |
| | CAPI (half sample) | Standardized and semi-standardized observational measures (incl. HLE) | July–Dec 2013 | 17 months | 1,510 | | | | 2015 |
| 3 | CAPI, PAPI | Semi-standardized observational measures (incl. HLE) | Apr–Nov 2014 | 25 months | 2,609 | 449 | | 110 | 2016 |
| 4 | CAPI, PAPI | Tabled-based competence tests | Apr–Nov 2015 | 3 years | 2,478 | 625 | 571 | | 2017 |
| 5 | CAPI, PAPI | Tabled-based competence tests | Apr–Sep 2016 | 4 years | 2,381 | 628 | 521 | | 2018 |
| 6 | CAPI, PAPI | Tabled-based competence tests | Mar–Aug 2017 | 5 years | 2,209 | 683 | 543 | | 2019 |
| 7 | CAPI, PAPI | Tabled-based competence tests | Apr–Sep 2018 | 6 years | 2,116 | 546 | 444 | | 2020 |
| 8 | CAPI, PAPI | Tabled-based competence tests | Mar–Aug 2019 | 7 years (mainly grade 1) | 2,070 | | | | 2021 |
| 9 | CATI-Remote[1]/ CAPI by phone[2,3] | Online Tests | June[3]–Sep 2020 | 8 years (mainly grade 2) | 1,848 | | | | 2022 |

**Table 1** Overview of all waves, used modes, target age of the child, participating families and childcare institution.

*Notes*: CAPI = Computer-assisted personal interview; CATI = Computer-assisted telephone interview; PAPI = Paper-and-pencil-Questionnaire; SUF = Scientific use file; HLE = Home learning environment; Each year, beginning with wave 3, there was also a small CATI –field for all families who did not take part in the CAPI. [1]Due to the Covid-19 pandemic, in wave 9 nearly all interviews with the parents were administered via telephone from the regular CAPI-interviewers from their homes. [2]The competence tests for the children were administered online with support of an interviewer by telephone. [3]The field of the original CAPI started in March 2020 before it stopped due to the pandemic. The field was re-opened in June 2020 with telephone interviews and the online testing.

Hence, across the waves, for 1,563 children of the SC1 information from early child care personnel are available (at least one questionnaire available from childminders, educators, or head of institution).

## 2.2 TIME OF DATA COLLECTION

The first assessment wave started in August 2012, the so-far latest assessment included in this paper took place in 2020 (assessments in the SC1 are still continued). Table 1 gives an overview of the duration of the field for each wave.

For further information see study overview https://www.neps-data.de/Data-Center/Data-and-Documentation/Start-Cohort-Newborns/Documentation.

## 2.3 LOCATION OF DATA COLLECTION

Data was collected at various sample points in Germany (see also 2.4.). In most waves, data collection took place in the households of the families.

## 2.4 SAMPLING, SAMPLE, AND DATA COLLECTION

The targets of SC1 are the children. As already mentioned, the first wave was intended to take place when the infants were 6 to 8 months of age. As context persons, one parent (respondent, in most cases the mother) as well as child care personnel were interviewed.

The sample was drawn via a register-based sample of addresses available at the level of municipalities. A two-stage disproportional stratified sampling strategy was used to allow for a representative sample of the newborn population in Germany. Overall, 84 German municipalities were considered as primary sampling units (Würbach et al., 2016). As a secondary step, addresses out of these municipalities were drawn. Overall, 8,483 addresses from 90 sampling points in 84 municipalities were used. A detailed description of the sampling strategy can be found in Würbach et al. (2016). Due to the fast developmental progress of infants

and young children and to achieve valid measurements of the included direct measures a strict age range for the sample of SC1 was intended. To achieve this goal, the drawn addresses of infants and their families were divided into two birth tranches (infants born between February and April 2012 and between May and July 2012). Starting from a gross sample of 8,483 families, a total of 3,481 families (response rate 41 %) took part in the first assessment wave. Of that, 3,431 gave their panel consent for participation in the following waves. Due to the high response rate, the addresses from infants born in July 2012 weren't contacted. Participation rate across the different waves was high with most waves arriving a participation rate of about 80% with a range between 79% and 87%. In wave 9 still 1,848 families[4] took part in the study. The description of the sample for each wave, including basic demographic information, is summarized in Table 2.[5]

As in all starting cohorts of the NEPS, participation in the SC1 is voluntary. As in most longitudinal surveys, also in the SC1 unit nonresponses and panel attrition occurred. To account for that and to allow to interpret analyses with the data of SC1 and draw conclusions regarding the population, design weights as well as weights to adjust for nonresponse are provided (see Würbach, 2017, 2018, 2019, 2020, 2021, 2022; Würbach et al., 2016; Würbach, Landrock, Schnapp, Ziesmer, & Bergrab, 2021).

In order to keep the families in the panel, the parents regularly received an incentive for participating in the interview (10€ per wave). In addition, the children received a small gift (value of 5€) as well as a certificate for participating. Further, to enhance panel stability, the families received summer and winter cards between waves.

## 2.5 MATERIALS/SURVEY INSTRUMENTS

In SC1 the following survey instruments were used: standardized and semi-standardized observation measures and competence tests for children as well as interviews and/or questionnaires for context persons, namely parents and external childcare personnel. An overview of the measured constructs with respect to child characteristics and development can be found in table A1 in the appendix. Constructs regarding the familial learning environment are compiled in table A2 and for institutional learning environment in table A3 in the appendix.

### Observational measures and standardized tests in SC1

In the NEPS, the following individual abilities and competencies are assessed: domain-general cognitive abilities/capacities as well as domain-specific cognitive competencies, metacompetencies as well as social competencies; and stage-specific skills (see Fuß, Gnambs, Lockl, & Attig, 2021 for an overview). In the following

a short description of the measures used in SC1 will be given as competence development is of special relevance to psychologists (see table A1 in the appendix for overview). A short description for each measurement point as well as technical reports for most of the competence measures can be found here: https://www. neps-data.de/Data-Center/Data-and-Documentation/ Start-Cohort-Newborns/Documentation ('Information on Competence tests'). Note that information on constructs such as the socio-emotional development or the temperament of the child are collected in the parent interview (see table A1 in the appendix for an overview).

### Observational measures (wave 1 to 3)

Due to the age of the children, in the first three waves observational measures were used.

In wave 1, four standardized tasks with toys (16 coded indicators in the scientific use file; SUF) were employed to measure facets of the sensorimotor development of the infants (see also Bayley, 2006). Due to the household setting, the tasks were video-taped and coded offline according to the intended behavior (i.e., whether the children showed the respective behaviour or not).

In wave 1 and 2, a visual habituation-dishabituation paradigm was implemented. In this paradigm, infant's visual attention is measured during the presentation of a series of stimuli presented by a computer. This procedure aims to measure the infant's ability to build a cognitive representation of the stimulus or the stimulus category (Pahnke, 2007). Such kind of measures have been shown to be predictive of intelligence scores or other indicators of cognition and language assessed later in life (Bornstein & Sigman, 1986; Fagan & Singer, 1983; Kavšek, 2004). In wave 1, two sets of stimuli were presented (both sets with focus on domain-general ability), in wave 2 three sets were presented[6] (one domain-general set already presented in wave 1; one with focus on language and one with focus on numeracy precursors; see Seitz, Attig, Möwisch, & Weinert, 2023 for details). In both waves, the children's looking behavior was videotaped during the presentation of the stimuli. Again, the videos were coded afterwards using Mangold INTERACT software. In this case, looking to the target was coded (with 30 frames per second).

In addition, a semi-standardized parent-child interaction setting was applied in wave 1 to 3 (adapted from the NICHD SECCYD study; NICHD Early Child Care Research Network, 1991). The interaction situations were also videotaped and coded afterwards by extensively trained coders. In the interactional situation, the parent was asked to play as natural as possible for 8 (wave 1; 5 minutes coded) or 10 minutes (wave 2 and 3) with toys provided from the NEPS. The interaction situations were coded with a macro analytic rating to parental (e.g. sensitivity, stimulation, emotionality) and child behavior (e.g. child's mood, activity level, sustained attention to

| WAVE | YEAR | PANEL COHORT (n)[1] | PARTICIPANTS: FAMILIES (n)[2] | PARTICIPATION RATE | AGE CHILD, MONTHS (m/sd) | SEX CHILD (FEMALE) | AGE RESPONDENT, YEARS (m/sd) | SEX RESPONDENT (FEMALE) | EDUCATIONAL BACKGROUND RESPONDENT, YEARS (m/sd) | HISEI, (m/sd) | MIGRATION |
|---|---|---|---|---|---|---|---|---|---|---|---|
| Wave 1 | 2012/2013 | – | 3,481 | 41% | 7.02 (0.76) | 49% | 32.37 (5.21) | 98% | 14.53 (2.65) | 60.20 (20.90) | 34% |
| Wave 2 | 2013 | 3,431 | 2,849 (CATI) 1,510 (CAPI) | 83% (CATI) 80% (CAPI) | 17.09 (0.61) | 49% | 33.34 (5.04) | 98% | 14.84 (2.54) | 62.55 (19.90) | 31% |
| Wave 3 | 2014 | 3,281 | 2,609 | 80% | 26.63 (1.22) | 49% | 34.61 (5.03) | 97% | 14.98 (2.48) | 59.81 (20.45) | 29% |
| Wave 4 | 2015 | 3,143 | 2,478 | 79% | 38.60 (1.08) | 49% | 35.64 (4.97) | 97% | 15.04 (2.48) | 58.68 (21.25) | 28% |
| Wave 5 | 2016 | 2,872 | 2,381 | 83% | 50.34 (1.70) | 49% | 36.75 (4.94) | 97% | 15.14 (2.43) | 59.10 (20.99) | 27% |
| Wave 6 | 2017 | 2,665 | 2,209 | 83% | 61.41 (1.80) | 50% | 37.72 (4.81) | 97% | 15.21 (2.39) | 59.34 (20.53) | 26% |
| Wave 7 | 2018 | 2,504 | 2,116 | 85% | 74.07 (1.60) | 50% | 38.85 (4.78) | 97% | 15.26 (2.37) | 59.59 (20.32) | 25% |
| Wave 8 | 2019 | 2,380 | 2,070 | 87% | 85.17 (1.79) | 50% | 39.85 (4.77) | 97% | 15.30 (2.39) | 59.93 (20.03) | 26% |
| Wave 9 | 2020 | 2,257 | 1,848 | 82% | 99.52 (1.56) | 49% | 41.05 (4.68) | 97% | 15.42 (2.29) | 60.65 (19.41) | 24% |

**Table 2** Sample description (child and respondent).

*Notes:* [1]Panel cohort = Number of families who agreed to take part in the NEPS. [2]Participants = Number of families that actually participated in the respective wave. Sex = Percentage of female children/interview respondents, Age (m, sd) = Mean age and standard deviation of children/interview respondents. Educational background (m, sd) = mean and standard deviation years of education of interview respondents. HISEI (m, sd) = Mean highest parental international socio-economic index of occupational status (Ganzeboom et al., 2010) and standard deviation. Migration = Percentage of families with interview respondent and/or partner born abroad.

objects, social interest) on five-point qualitatively defined rating scales (see Linberg et al., 2019 for a more detailed description of the situation and the coding).

*Tablet-based standardized tests*

Beginning with age 3 (wave 4), tablet-based standardized tests of children's abilities, skills, and competencies were used (see also Weinert et al., 2019). In all waves, the interviewer administered the test by a tablet and the child was asked to indicate her/his answer with a touch on the tablet screen (or with an additional keyboard in the executive functions measures).

In wave 4, 6 and 8 the receptive vocabulary (majority language – German) of the children was measured using a German Version of the Peabody Picture Vocabulary Test (PPVT-IV; Dunn & Dunn, 2007; German version by Lenhard, Lenhard, Segerer, & Suggate, 2015). The PPVT is an international well-known test to measure receptive vocabulary across different age groups (Dunn, 1959; Dunn & Dunn, 1981, 1997, 2007). The children were presented with a spoken word and four pictures and had to match the word to corresponding one of the four pictures.

In wave 4 and 7, a digit span task (based on the German version of the "Kaufman Assessment Battery for Children"; K-ABC; Melchers & Preuß, 2009) was administered to measure the capacity of phonological working memory. In span tasks, sequences of digits (with an increasing number of digits) are presented auditorily and the child is asked to repeat each sequence immediately in the correct order. In wave 8, a backward digit span tasks was used. In this case, the children had to repeat the heard digits in reverse order (K-ABC; Melchers & Preuß, 2009).

To obtain a brief indicator of non-verbal basic cognitive abilities, the subtest "Categories" of the Snijders-Oomen Nonverbal Intelligence Test (SON-R 2½–7; Tellegen, Laros, & Petermann, 2007) was conducted in wave 4. The children's task is to sort carts according to specific characteristics. In wave 7, children's basic cognitive skills were indicated by two different tasks/components: perceptual speed and reasoning (Autorenteam Kompetenzsäule, 2020). Perceptual speed was measured by a Picture Symbol task (NEPS-BZT), where children had to enter the correct figures for the presented symbols. The NEPS reasoning test (NEPS-MAT) is a matrices test; again the children had to select the correct element from different presented elements that fits into a given figural matrix.

Further, in wave 4, 6 and 8 a delay of gratification task was included to measure inhibitory control ("hot" executive functions). In wave 4 and 6, a waiting paradigm was used to examine children's ability to wait for a bigger reward (Mischel & Gilligan, 1964). The children were presented with one small and one big gift and a USB button was placed between the gifts. The children were told to wait for a timespan unknown to

them to receive the big gift. Children who didn't want to wait (further) pressed the button and received the small gift. The children who waited received the big gift. In wave 4, the waiting time was three minutes, in wave 6 five minutes. In wave 6, an additional rating of child behavior during the waiting time was integrated. In wave 8, a choice paradigm was included. The children had to choose between two options: receiving one gift directly or receiving two gifts tomorrow. Hence, the children had to make a decision.

In wave 5, again components of executive functions were measured, this time with an age-appropriate flanker task. With flanker tasks, different components can be assessed: inhibitory control, selective attention, and cognitive flexibility (if a rule change is included). In the SC1, the stimulus material was presented via tablet computer which was connected with a reaction time keyboard. As directional indicators fishes were shown. The children were presented with the pictures of a school of fishes and had to press a button to indicate the direction of the middle fish. To switch rules, in a second task they had to focus on the outer fishes. As usual in flanker tasks, there are congruent (all fishes swim in the same direction) and incongruent trials (the middle fish and the outer fishes swim in different directions).

In wave 5, 7, and 9 mathematical competence of the children was assessed. All mathematical tests in the NEPS are based on the concept of mathematical literacy and were developed for the NEPS. The test items involve different content areas as well as different cognitive components. More information can be found in the respective technical reports (Autorenteam Kompetenzsäule, 2020; Kock, Litteck, & Petersen, 2020; Petersen, Beyer, & Bednorz, 2022; Petersen & Gerken, 2021; Weinert et al., 2019).

In wave 6 and 8 scientific literacy was measured (see Autorenteam Kompetenzsäule, 2020; Weinert et al., 2019). Again, the tests were developed especially for the NEPS. Similar to the mathematical tests, they involve different content areas and processes and are designed to assess the children's scientific literacy/knowledge in environmental, technological, and health contexts (Hahn et al., 2013). For more information see the Technical reports from Hahn (2019, 2021).

In wave 9, reading speed and early reading comprehension were tested (see Autorenteam Kompetenzsäule, 2020; Weinert et al., 2019). To assess reading speed the Salzburg Reading Screening for Grades 2–9 (Wimmer & Mayringer, 2014) was conducted. The children got a simple sentence and were asked to indicate whether the sentence was meaningful or not. Due to the pandemic, the test was mainly administered online, hence, the children could give their answers on touchscreen or keyboard. Reading speed is measured by the number of sentences judged correctly within a given time interval. To measure early reading comprehension

the ELFE II test (a Reading Comprehension Test for First to Seventh Graders; Lenhard, Lenhard, & Schneider, 2017) was deployed. The children had to read short texts and answer questions to the texts.

Note that the NEPS provides longitudinally linked WLE for the mathematical and the science competence of the children. The PPVT-scores can also be compared directly across measurement points.

### Interview and questionnaire data in the SC1
*Information on child characteristics and learning environment in the family*
In the interview with the parent, the child's biographical key data is collected alongside with information on various behavioural, health and developmental features of the child (see table A1 in the appendix for an overview). Beyond that, different aspects of the learning environment of the family are covered. On the one hand, more distal background information such as the sociodemographic/ structural characteristics of the family were collected. On the other hand, information to educationally relevant knowledge and processes as well as to mental orientations such as opinions, attitudes, beliefs and aspirations were asked. Last, also information about health and personality characteristics of the respondent were covered (see table A2 in the appendix for an overview).

*Information on the institutional learning environment*
Data is collected from the parents as well as from the institutions themselves (ECCE/preschool). Following the idea above, information on structure, process and orientation characteristics as well as further information are contained (see table A3 in the appendix for an overview).

The interview questions and the paper-and-pencil questionnaires for parents and child care personnel are published and can be found at: https://www.neps-data. de/Data-Center/Data-and-Documentation/Start-Cohort-Newborns/Documentation ('Survey Instruments'). As an exception, one questionnaire (the early child language checklist; ELFRA-2 (Grimm & Doil, 2006)) isn't included in this NEPS documentation as it is a published instrument whose items are copyrighted. This checklist is comparable to the well-known "MacArthur-Bates Communicative Development Inventories" (CDI; Fenson et al., 1993) and was presented to the parents in wave 3. Besides 260 words as vocabulary checklist, the ELFRA-2 also includes 26 items on child's syntax and 11 items on morphological aspects (for the validity of the ELFRA, see e.g., Sachse, Anke, & Suchodoletz, 2007).

### 2.6 QUALITY CONTROL
Especially in the first waves, innovative observational measures for a longitudinal large-scale panel study which visits the families at their homes were used. As such, different steps were done to ensure a high quality

of the assessments. Amongst others, a longitudinal pilot study was conducted, with assessments taking place usually one year before the main study. In the pilot study, all constructs and instruments were tested. The pilot study also enabled to check the administration as well as the training of the interviewers for the main survey. Furthermore, all instruments/items that were to be included in the final main studies' questionnaires/ interviews had to be checked for existing evidence on their reliability and validity (e.g. in case of already existing scales or those developed or used in other large-scale studies) or to be pretested in qualitative cognitive interviews or tested in special studies on instrument development. Further, with respect to the SC1 an additional validation project (funded by the German Research Foundation within the Priority Programme 1646) was set up to validate included short scales of constructs and the coding and stability of observations (see e.g., Weinert et al., 2023).

In addition, the following steps were included in the main survey to ensure the quality of the administration of instruments and measurements:

All interviewers had to take part in an extensive training before start of the field work. For each assessment wave, interviewers had to pass a training consisting of two blocks, each lasting two days, respectively. Besides information on the study, the participating families, and the used instruments, the training also included work in small groups to practice the administration of the observational and competence measures. To get the permission to collect data in the field, each interviewer had to record a video proving the correct administration of the standardized tasks/competence tests. When problems in the administration were identified, the interviewers were retrained or finally excluded from the field work if they weren't able to meet the necessary quality standards. In addition, during data collection of the first three waves, all assessments in the field were videotaped. These videos were rated concerning the correct administration of measurements. If errors occurred, the interviewers were trained again or, if they did not meet the criteria, finally excluded from the field work. Beginning with wave 4, tablet-based measures were used to facilitate the standardization and administration of assessments. As a field monitoring measure, selected interviews were still videotaped to identify incorrect administrations, e.g., of the competence tests. In addition, quality checks were also implemented with respect to the household and telephone interviews. In particular, audio recordings were made and controlled afterwards. The records did not only show whether the interviewer administered all components of the interview correctly and appropriately, but also showed whether the respondents understood the

questions correctly or whether they had problems with some of them (e.g., had queries). This information was used to improve the survey in the following waves. Last but not least, the NEPS also used the opportunity to accompany the interviewers in their work to get an impression how the survey is conducted and works in the households.

## 2.7 DATA ANONYMISATION AND ETHICAL ISSUES

The NEPS study is conducted under the supervision of the German Federal Commissioner for Data Protection and Freedom of Information (BfDI) and in coordination with the German Standing Conference of the Ministers of Education and Cultural Affairs (KMK) and – in the case of surveys at schools – the Educational Ministries of the respective Federal States. All data collection procedures, instruments and documents were checked by the data protection unit of the Leibniz Institute for Educational Trajectories (LIfBi). The necessary steps are taken to protect participants' confidentiality according to national and international regulations of data security. Participation in the NEPS study is voluntary and based on the informed consent of participants. This consent to participate in the NEPS study can be revoked at any time. All parents of the SC1 of the NEPS give their agreement for participation and answering questions during the assessments as well as a written consent for participating in the video-taped measures at each measurement point.

In NEPS SC1 all collected data come from individuals. Hence, before the SUF were published, strict international standards to ensure confidentiality protection of individuals and their data were applied. Identifiable information in the data is thrown out before publishing the data. All information to the anonymization process concerning SC1 can be found in Koberg (2022), the paper included a list of all affected variables and examples. For example, for the current SUF (Version 9.1.0) 222 out of 5995 variables were affected from the anonymization processes.

Please note that all information coming from the institutional care is only available Onsite or in RemoteNEPS to meet the data protection issues.

## 2.8 EXISTING USE OF DATA

The data collection of the NEPS was and still is intended for secondary analyses by the scientific community.

More information can be found at https://www.neps-data.de/Project-Overview/Publications and https://www.neps-data.de/Data-Center/Research-Projects.

In the following, some recent examples of publications using NEPS SC1 data are given:

Attig, M., & Weinert, S. (2020). What impacts early language skills? Effects of social disparities and different

process characteristics of the home learning environment in the first two years. *Frontiers in Psychology, 11,* Article 557751. doi: https://doi.org/10.3389/fpsyg.2020.557751

Freund, J.-D., Linberg, A., & Weinert, S. (2019). Longitudinal interplay of young children's negative affectivity and maternal interaction quality in the context of unequal psychosocial resources. *Infant Behavior and Development, 55,* 123–132. doi: https://doi.org/10.1016/j.infbeh.2019.01.003

Huang, W., Weinert, S., Wareham, H., Law, J., Attig, M., von Maurice, J., & Rossbach, H.-G. (2022). The emergence of 5-year-olds' behavioral difficulties: Analyzing risk and protective pathways in the United Kingdom and Germany. *Frontiers in Psychology, 12,* Article 769057. doi: https://doi.org/10.3389/fpsyg.2021.769057

Konrad-Ristau, K., & Burghardt, L. (2021). Differences in children's social development: How migration background impacts the effect of early institutional childcare upon children's prosocial behavior and peer problems. *Frontiers in Psychology, 12,* Article 614844. doi: https://doi.org/10.3389/fpsyg.2021.614844

Linberg, A., Attig, M., & Weinert, S. (2020). Social disparities in the vocabulary of 2-year-old children and the mediating effect of language-stimulating interaction behavior. *Journal for Educational Research Online, 12,* 12–35. doi: https://doi.org/10.25656/01:20971

Linberg, A., Lehrl, S., & Weinert, S. (2020). The early years home learning environment – Associations with parent-child-course attendance and children's vocabulary at age 3. *Frontiers in Psychology, 11,* Article 1425. doi: https://doi.org/10.3389/fpsyg.2020.01425

Möwisch, D., Konrad-Ristau, K., & Weinert, S. (2022). Cognitively stimulating maternal language as predictor for vocabulary growth. *Zeitschrift für Erziehungswissenschaft, 26,* 319–344. doi: https://doi.org/10.1007/s11618-022-01114-y

Novita, S., & Kluczniok, K. (2021). Receptive vocabulary of preschool children with migration backgrounds: The effect of home literacy activities. *Early Child Development and Care, 192*(11), 1728–1743. doi: https://doi.org/10.1080/03004430.2021.1932861

Oeri, N., & Roebers, C. M. (2022). Adversity in early childhood: Long-term effects on early academic skills. *Child Abuse & Neglect, 125,* Article 105507. doi: https://doi.org/10.1016/j.chiabu.2022.105507

Seitz, M., & Weinert, S. (2022). Numeracy skills in young children as predictors of mathematical competence. *British Journal of Developmental Psychology, 40*(2), 224–241. doi: https://doi.org/10.1111/bjdp.12408

## (3) DATASET DESCRIPTION AND ACCESS

The collected data in NEPS are meant to be published for secondary analyses. As such, the data of the NEPS SC1 is edited in a user-friendly way to the scientific community.

The preparation of the SUF is mostly done at the Research Data Center of the LIfBi (FDZ-LIfBi, 2022). The Research Data Center has received accreditation from the German Data Forum (RatSWD) related to uniform and transparent standards. As all data as well as most of the documentation are available in German and English, NEPS data can be used (free of charge) by national as well as international researchers.

There are three ways to access the NEPS data. This ensures maximum data usability while complying with strict data protection regulations. For data access interested researchers are obliged to submit a data use agreement. The three access modes are linked to three data versions which differ in their degree of data anonymization and data security provisions:

1. Download SUF: in this data version the strongest anonymization is implemented in comparison to the Remote or On-site versions. In the Download version sensitive information is recoded or removed. Interested researchers must submit a data use agreement and can then download the SUF from the NEPS website (for further information see chapter 3.1)
2. Remote: this data version has a moderate level of anonymization. Additionally, as the Remote data version can only be accessed and analyzed in the controlled RemoteNEPS web environment, a RemoteNEPS supplemental agreement and a registration of keystroke biometrics for authentication is required. The RemoteNEPS serves as a "virtual desktop" for the user and provides all infrastructure necessary for data processing and analysis. No software installation is required, the use of statistic programs in RemoteNEPS is free of charge and the setup of shared folders for collaborative working is possible. Technically, the use of the Remote NEPS only requires web access and a current browser.
3. On-site: this data version contains the lowest level of anonymization and includes also sensitive data. Interested users have to submit a data use agreement and an additional On-site supplemental agreement. Access to On-site data is only possible at the LIfBi in Bamberg, where workstations in data protection rooms are provided. These data protection rooms are disconnected from the internet and all input to and output from the protected environment is controlled.

All required date use and additional agreement forms as well as an overview on sensitive information provided in the respective dataset versions and online access for authorized users to the SUF download area and RemoteNEPS environment is available at: https://www.neps-data.de/Data-Center/Data-Access.

## 3.1 REPOSITORY LOCATION
SC1 is a longitudinal study. Hence, nearly every year a new SUF is published and labeled with a Digital Object Identifier (DOI). The following information provides the DOI: amount of waves, survey timeframes as well as amount of larger and smaller updates.

The doi of the current SUF of the data of the SC1 is https://doi.org/10.5157/NEPS:SC1:9.1.0

Note that a new SUF includes – in addition to the newly published data – all data of the previous SUF.

## 3.2 OBJECT/FILE NAME
In the SUF of SC1 multiple data sets can be found. As the NEPS is conceptualized by an interdisciplinary network, cross-sectional, longitudinal and episode or spell data sets are integrated from the different assessments in SC1 forming a complex data structure. This is reflected in the data structure of the available data sets. While longitudinal data is provided usually in long data format, episodes, like e.g., in childcare or respondent's employment, are available as spell data, where one case represents one episode of one respondent. The ID of the child can be used to merge the different data sets.

An overview of available data sets in the current SUF can be found in Table 3.

## 3.3 DATA TYPE
The data of SC1 SUF comprises primary and processed data and is made available to the scientific public for secondary data analyses.

## 3.4 FORMAT NAMES AND VERSIONS
All data sets are available for the statistical software packages SPSS (.sav) and Stata (.dta).

## 3.5 LANGUAGE
The data sets and documentation are available in German and English. Further, the website as well as the data use agreements are also available in German and English.

## 3.6 LICENSE
SC1 as the whole NEPS study is carried out by a nation-wide interdisciplinary scientific network of researchers. It covers a broad spectrum of topics and is intended to address researchers from various different disciplines. To meet the high standards of analyses, only the scientific community gets access to the data. Commercial or other economic purpose is not permitted. The use of the data is free of charge and LIfBi offers different access possibilities as well as access to the statistical software to analyze the data in RemoteNEPS and the data security rooms of the LIfBi. To get access to the NEPS data, researchers need the connection to a research institution and have to conclude a data use agreement with the LIfBi. With

| NAME | SHORT DESCRIPTION | FILE STRUCTURE | ACCESS (ON-SITE, REMOTE, DOWNLOAD) |
|------|------------------|----------------|-----------------------------------|
| SC1_CohortProfile | Basic information on target persons in initial cohort panel sample | long | Download Remote On-site |
| SC1_EditonBackups | Generated file; original data on values changed in data edition process | long | Download Remote On-site |
| SC1_Weights | Generated file; sample weights | wide | Download Remote On-site |
| SC1_pParentMicrom | Small scale regional indicators and respondent's place of residence | panel format; long | On-site |
| SC1_xTargetCompetencies | Competencies: domain and stage specific | wide | Download Remote On-site |
| SC1_xPlausibleValues | Competencies: measurement error adjusted for group level description | wide | Download Remote On-site |
| SC1_xDirectMeasures | Competencies: coded from observation at home | wide | Download Remote On-site |
| SC1_MethodsDirectMeasures | Competencies: information on data collection of direct measures | long | Download Remote On-site |
| SC1_pEducator | Context Person: questionnaire data from educators in day-care institutions | long | Remote On-site |
| SC1_pEducatorChildminder | Context Person: questionnaire data from childminders | long | Remote On-site |
| SC1_pInstitution | Context Person: questionnaire data from heads of institutions | long | Remote On-site |
| SC1_pParent | Context Person: interview data from parent (usually mother) | long | Download Remote On-site |
| SC1_pParentCORONA* | Context Person: interview data from parent on the COVID-19 pandemic | | Download Remote On-site |
| SC1_MethodsCAPI | Context Person: information on data collection during parents' CAPI | long | Download Remote On-site |
| SC1_MethodsCATI | Context Person: information on data collection during parents' CATI | long | Download Remote On-site |
| SC1_spSibling | Entity spells: information on siblings of target person | entity format: 1row = 1 sibling of 1 respondent | Download Remote On-site |
| SC1_spChildCare | Episode spells: information on all child care episodes | spell | Download Remote On-site |
| SC1_spEmp | Episode spells: information on parent's employment episodes | spell | Download Remote On-site |
| SC1_spPartnerEmp | Episode spells: information on parent's partner's employment episodes | spell | Download Remote On-site |

(Contd.)

| NAME | SHORT DESCRIPTION | FILE STRUCTURE | ACCESS (ON-SITE, REMOTE, DOWNLOAD) |
|---|---|---|---|
| SC1_spParLeave | Episode spells: information on parent's parental leave episodes | spell | Download Remote On-site |
| SC1_spPartnerParLeave | Episode spells: information on parent's partner's parental leave episodes | spell | Download Remote On-site |
| SC1_spParentGap | Episode spells: gap episodes in target child's lifecourse | spell | Download Remote On-site |
| SC1_psParentSchool | Episode spells: episodes of target child's schooling | spell | Download Remote On-site |

**Table 3** Overview of available data sets of the NEPS SC1 (version NEPS:SC1:9.1.0).

*Note*: Prefixes of the data sets: x = cross-sectional structure, p = panel structure, sp = spell file;
Suffixes of the data set: _O = On-site, _R = Remote, _D = Download; wave, major and minor Updates (e.g. current SUF = _9-1-0).

the agreement researcher get access to the data for the whole project time. The LIfBi as well as the network of the NEPS are committed to Open Science taking data protection as well the protection of the rights of the participants into account.

### 3.7 LIMITS TO SHARING

As mentioned above, access to the NEPS data requires the conclusion of a data use agreement with the LIfBi.

The NEPS provides three different types of data access which depend on the level of data protection and anonymization (see chapter 3). To meet the high standard of data protection, there is also data which is not published in the SUF. All videos recorded in wave 1 to 3 from the three observational measures are not published to provide confidentiality protection of participating families' identity. Nevertheless, coding of these observational measures are made available in the SUF.

### 3.8 PUBLICATION DATE

In Table 1 the publication year of the respective SUF can be found. The first SUF with data of SC1 was published in 2015, the current SUF (version NEPS:SC1:9.1.0) was published 2022.

### 3.9 FAIR DATA/CODEBOOK

As the data of NEPS SC1 is published as SUF and intended to be used for secondary analyses from national and international researchers, all collected data are thoroughly prepared, documented, disseminated and free of charge for users.

NEPS data comply to the FAIR guidelines (Findability, Accessibility, Interoperability, and Reuse). The data of SC1 can be found on the website of the NEPS (https://www.neps-data.de/Data-Center/Data-and-Documentation/Start-Cohort-Newborns; findability). Further, all researchers working with the data have to cite the NEPS study. To introduce NEPS to researchers,

presentations at conferences, workshops, trainings as well as newsletters are offered to spread the potential of the data. NEPS provides three different types of data access (accessibility; see chapter 3 for more information). The types differ in the amount of anonymization. To get access to the data, researchers need a data use agreement. In all three types, the NEPS data are provided for different versions of SPSS (sav) and STATA (dta) with the respective metadata (interoperability). Publishing the data only as sav and dta might be a limitation for all researchers who don't have access to these statistical programs. As the NEPS provides access to different statistical software packages (SPSS, R, Stata, Mplus) in the RemoteNEPS environment, researchers can work there to analyze the NEPS data. The meta-data can also be found in the NEPSplorer (https://www.neps-data.de/Data-Center/Overview-and-Assistance/NEPSplorer). Further, information on the content of all data sets are accessible in the codebook of each version of the SUF (see here https://www.neps-data.de/Data-Center/Data-and-Documentation/Start-Cohort-Newborns/Documentation 'Codebook'). Interview and questionnaire items are documented and accessible without any contract. As already mentioned, the data of SC1 in the NEPS (as the data of all other NEPS starting cohorts) are generated for secondary analyses (reuse). The FDZ also offers training courses several times a year to introduce the datasets and give hints how to work with them. Even researchers developing and preparing the data collection of the SC1 have to wait for the published SUF to analyze the data (only exception: analyses relevant to data quality).

## (4) REUSE POTENTIAL

The NEPS study, and with it the SC1, was initiated to collect longitudinal data on educational trajectories and competence development across the life span.

By now, the SC1 provides a broad data base for analyzing the conditions that significantly contribute to early developmental progress in the first eight years. The first assessment with infants and their families took place in the first year of the children's life and the children and their families are now accompanied over several years. As such, the data sets of SC1 contain extensive information on different longitudinally measured skills, abilities and competencies of the children including measures of cognitive abilities in the first year of life. The competence data is flanked by a wide spectrum of further characteristics of the children (e.g., ratings of child behavior, personality, and socio-emotional development) as well as information on the families and the learning environments the children grow up in conceived from the perspective of various scientific disciplines. This comprises direct observations of the quality of the interaction behavior of the parent and the child in the first two years as well as information on further learning environments of the child – the institutional care setting and comprehensive information on socioeconomic, education- and migration-related background variables and capitals in the family. Even in comparison to other international cohort studies (e.g., ECLS-B[7]; Millennium Cohort[8]), the SC1 offers the following unique features: (1) direct measures of basic cognitive skills of the infants from the first wave on (infant age 6 to 8 months), (2) three measurement points with direct measures in the first two years of life and annual assessments thereafter often allowing for modeling individual change across time, (3) tablet-based standardized tests beginning with 3-year-old children.

Thus, especially for a large-scale longitudinal survey, the SC1 is closely accompanying the development of competencies of the children and a broad range of influential factors. Hence, this rich data pool offers a broad potential of analyses of the educational trajectories from the crib on. It can be used to show whether and how early outcomes of the children are the basis for future developments and on the role of learning environments and social background variables. The data allows the analysis of influencing conditions, factors and educational processes that are important for explaining differences in the development of the children. As longitudinal coherently assessed measures of competencies are available, complex growth-curve modeling and the modeling of intra-individual change are possible which enables researchers to address the effects of various factors on developmental trajectories. The age range of the children investigated stretches from the first years up to the transition to primary school and even beyond primary school.

In the following we give some examples of the broad range of research questions which can be addressed and analyzed with the help of the data of SC1:

– How do competencies and skills that are relevant for education develop in the first years of life? How do they predict later competencies and school success? How are different domains of development longitudinally related to each other?

– To what extend is the development of educationally relevant competencies influenced by the learning environments of children and how do they interact? Which role do child characteristics and competencies play in the decision of the parents for the use and starting point of institutional care? What are important process characteristics in the family for the development of child competencies and skills and do they change over the first years of life? Do all children benefit from the same process characteristics of the home learning environment? Or with other words: What are specific risk (or protective) factors and potentially compensatory factors? When do social or ethnical disparities in child competencies and skills emerge and how do they develop over time? Which aspects of the different learning environments impact on these disparities in that they increase or help to decrease these disparities and help children to overcome such disadvantages?

– At what age do the children enter institutional care and what are the influencing factors on this decision? How are competencies and child characteristics influenced by the structure, learning processes, and, e.g., the point of time of entrance in preschool?

The purpose of the data use of NEPS data is bound on fundamental and applied scientific analyses. Insights and conclusion drawn from research with this data will be relevant for educational facilities and social policy and thus for the individual child as well as for society. As large-scale study with a representatively drawn sample, the data of the SC1 provide a high level of ecological validity that allows for generalization of research results. And even if the levels of standardization of the data collection process cannot correspond to those of experimental settings, the standardization requirements of the SC1 are high for a field study.

Analyzing the data of NEPS SC1 can produce crucial knowledge on how children's abilities, skills, and competencies develop based on individual resources and environmental conditions; how learning opportunities influence their development in different contexts; how disparities emerge early in life; and how all this impacts educational careers, lifelong learning, well-being, and participation in society. The SC1, as the whole NEPS, has a carefully drawn sample, the ecological validity of the data is high and the collected longitudinal data is published for secondary analyzes. The different NEPS cohorts allow for cohort comparisons. In addition, there is connectivity with international studies, allowing for

international comparisons (see for example Huang et al., 2022; Volodina et al., 2022).

## NOTES

1  The BiKS-3-18 study on educational processes, competence development, and formation of educational decisions started in 2005 with a sample of about 550 children at the age of three years tracing their development and relevant contextual factors across 15 years by extensive observations in the children's homes, in preschools, and later in schools, as well as by comprehensive tests, interviews, and questionnaires. Data is available as scientific use file (see IQB – https://www.iqb.hu-berlin.de/fdz/studies/BiKS_3-10). Further waves will be available in the near future.

2  Note that beginning with wave 10, the assessments with the children included not only the competence measures but also an interview.

3  To reduce the costs of the survey and the burden on the families, the second wave was planned as a CATI field. However, as early direct measures are rare in panel studies and offer high potential for research questions, a design was developed where at least about half of the panel sample was randomly selected for an additional assessment of observational measures in the households of the families. Thus, a random sample of 34 municipalities was drawn from the initial 84 municipalities (see section 2.4.). This subsample was asked to additionally participate in the household field.

4  Note that during wave 9, the first Corona-lockdown took place. Hence, the field has been stopped and restarted later on. Participation rate in this year was slightly smaller than in the waves before.

5  This paper uses data from the National Educational Panel Study (NEPS; see Blossfeld & Rossbach, 2019; NEPS Network, 2022). The NEPS is carried out by the Leibniz Institute for Educational Trajectories (LIfBi, Germany) in cooperation with a nationwide network.

6  The codings of one set in wave 1 and two sets in wave 2 are not integrated in the SUF and will be published soon.

7  https://nces.ed.gov/ecls/birth.asp.

8  https://cls.ucl.ac.uk/cls-studies/millennium-cohort-study/.

## SPECIAL COLLECTION

This submission is intended to be part of the Special Collection of data papers related to data for psychological research in the educational field, edited by Sonja Bayer, Katarina Blask, Timo Gnambs, Malte Jansen, Débora Maehler, Alexia Meyermann and Claudia Neuendorf.

## ADDITIONAL FILE

The additional file for this article can be found as follows:

- **Appendix.** Overview of the measured construct in NEPS-SC1. DOI: https://doi.org/10.5334/jopd.81.s1

## ACKNOWLEDGEMENTS

The authors like to thank Daniel Fuß who give advice to some sections of the manuscript (license, limits to sharing and fair data).

Further, the authors like to thank all families who participated in the NEPS.

## FUNDING INFORMATION

From 2008 to 2013, NEPS data was collected as part of the Framework Programme for the Promotion of Empirical Educational Research funded by the German Federal Ministry of Education and Research (BMBF). As of 2014, NEPS is carried out by the Leibniz Institute for Educational Trajectories (LlfBi, Germany) in cooperation with a nationwide network.

The publication of this article was funded by the Open Access Fund of the Leibniz Association.

## COMPETING INTERESTS

The authors have no competing interests to declare.

## AUTHOR CONTRIBUTIONS

MA wrote the first draft of the manuscript. MV prepared the tables with the measurements and constructs of the SC1 and improved the manuscript. SW revised the manuscript critically to improve the draft and contributed to all sections of the papers. All authors contribute to finalize the manuscript and approved the submitted version.

The NEPs is carried out by a nation-wide interdisciplinary scientific network of researchers. The NEPS associated staff of the LIfBi as well as of the NEPS network contributed to the assessments of the NEPS.

## AUTHOR AFFILIATIONS

**Manja Attig** orcid.org/0009-0007-4568-808X
Leibniz Institute for Educational Trajectories, DE

**Markus Vogelbacher**
Leibniz Institute for Educational Trajectories, DE

**Sabine Weinert** orcid.org/0000-0002-8341-9821
Universitity of Bamberg, DE

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

## PEER REVIEW COMMENTS

*Journal of Open Psychology Data* has blind peer review, which is unblinded upon article acceptance. The editorial history of this article can be downloaded here:

- **PR File 1.** Peer Review History. DOI: https://doi.org/10.5334/jopd.81.pr1

Attig et al. *Journal of Open Psychology Data* DOI: 10.5334/jopd.81

**TO CITE THIS ARTICLE:**

Attig, M., Vogelbacher, M., & Weinert, S. (2023). Education from the crib on: The potential of the Newborn Cohort of the German National Educational Panel Study. *Journal of Open Psychology Data,* 11: 13, pp. 1–18. DOI: https://doi.org/10.5334/jopd.81

**Submitted:** 30 September 2022     **Accepted:** 20 July 2023     **Published:** 03 August 2023

