## [Peer Review History. · Journal of Open Psychology Data]

LifBi · Wilhelmsplatz 3 · 96047 Bamberg · Germany

Dr. Debora B. Maehler
Editor in Chief
JOPD special collection “Data for Psychological Research
in the Educational field”

Bamberg, 11.07.2023

Responses to reviewers comments – JOPD – “Education from the crib on: The potential of the Newborn Cohort of the German National Educational Panel Study”

Dear Dr. Maehler,

thank you and the two reviewers for the carefully reading of our manuscript “Education from the crib on: The potential of the Newborn Cohort of the German National Educational Panel Study”.

We have carefully revised the manuscript based on the reviews and respond below to each suggestion. We appreciate the effort and helpful feedback of the reviewers.

We thank you for your consideration of our revised manuscript.

Kind regards

The Authors

Review 1

#1. Reviewer: Abstract.

Overall, this is a concisely written summary of the data paper. Perhaps the authors could think about mentioning the RDC at LfBi, which makes the data available for reuse, instead of just writing that it is accessible.

Perhaps one small thing, not really related to the abstract, but the manuscript as a whole: The authors should add a running head.

Response by authors:

Thanks a lot for the kind words. We added both suggestions in the manuscript.

Abstract:

“The data is accessible via the Research Data Center of the LfBi and comprehensively documented (English, German) to be used,.....”

Running head:

“Newborn Cohort of the NEPS”

#2. Reviewer: Background:

I really liked the way the authors present the NEPS study and the SC1 cohort. The introduction is very well structured and invites the reader to learn more about this really interesting data corpus.

Response by authors:

Thanks a lot for this kind summary of our text.

#3. Reviewer: Methods

1. Regarding table 1 I wondered whether readability might be increased by inverting the table so that it follows a similar structure as table 2 (column headers would then be: waves, interviewmode, HLE, Year, Child Age...). Moreover, there should be added a table header as for the other tables. Finally, I was a little bit confused by the information provided on the age of the children as it deviates from the information provided in table 2. I suppose that it is the median age in table 1 and the mean age in table 2. However, perhaps it would make sense to include the age information only in one table, in one format.

Response by authors:

Thanks for the suggestions. We inverted the table as suggested and added a header.

And we added “child target age” in table 1. So the difference between the two tables is the intended age for each wave and in table 2 the age after collecting the data. We hope that is now clear.

2. In section 2.1 in the last sentence of the first paragraph there is an unnecessary bracket before “PAPIs for educational...”

Response by authors:

Thanks for the really careful reading of the manuscript. We removed the bracket. In addition, we have reviewed the whole manuscript to hopefully correct all errors.

3. In section 2.1 the authors state that for wave 2 only half of the sample was surveyed in the children’s home. Perhaps the authors could be a little bit more precise regarding the mentioned design decisions.

Response by authors:

Yes of course. We added the following footnote to explain it more:

“To reduce the costs of the survey and the burden on the families, the second wave was planned as a CATI field. However, as early direct measures are rare in panel studies and offer high potential for research questions, a design was developed where at least about half of the panel sample was randomly selected for an additional assessment of observational measures in the households of the families. Thus, a random sample of 34 municipalities was drawn from the initial 84 municipalities (see section 2.4.). This subsample was asked to additionally participate in the household field.”

4. On p.7 the authors write that “interviewers had to pass a training consisting of 2 blocks”. I would recommend to write the number out.

Response by authors:

Thanks again. We changed it.

5. The description of section 2.5 is quite extensive with more than ten pages and also compared to the rest of the manuscript. It would be good if the authors reduced this. Perhaps the extensive information on the constructs, which cannot actually be directly related to the data, can be integrated into the description of the measures and tests as examples. Alternatively, the authors may limit the construct description to the child characteristics and include links/references to publications that provide further

information on the constructs presented in Tables 4 and 5. In any case, I would shorten this section to three or four pages.

Response by authors:

We checked the chapter carefully and added the following changes:

First, we removed the three tables from the chapter and added an appendix where the reader can now find the tables. Second, we also made small changes in the description of the observational measures and competence measures. Third, we added some information about the measured constructs in the interviews.

6. On p.18 an opening bracket before “CD; Fenson et al., 1993)” is missing.

7. p.19. 3rd line: Here a space is missing between “2” and “wave”.

8. p.19 in the paragraph on tablet-based standardized tests, second paragraph: Should be “children was measured using a German version of the Peabody Picture Vocabulary Test”

Response by authors:

Again, thanks for the carefully reading. We changed all the mentioned points.

9. The authors should ensure consistent formatting of all references listed in section 2.8, especially in relation to DOIs.

Response by authors:

Yes of course and thanks for the hint. We checked section 2.8. as well as the reference lists and corrected the errors.

#4. Reviewer: Dataset description and access

1. In Table 6 the information on the version of the SC1 dataset could be added to the table header.

Response by authors:

Ok, we added the version to the header as suggested.

“Table 3. Overview of available data sets of the NEPS SC1 (version NEPS:SC1:9.1.0)”

Note that it’s now table 3 (as we delete the three tables in section 2.5.).

2. In section 3.4 the authors write that all datasets are available for SPSS and Stata, but not in any non-proprietary format. So the data can only be used by researchers having access to SPSS or Stata. This is a limitation in terms of FAIRness, especially reusability, of the data and should be addressed in the paper.

Response by authors:

Thanks for the hint.

Even if the NEPS only publish the datasets for Stata and SPSS, all users can use the RemoteNEPS environment to work with the NEPS data and there the NEPS provide access SPSS; Stata, MPLus as well as R (free for all users).

We changed the part a little bit in the manuscript to make that more clear.

“In all three types, the NEPS data are provided for different version of SPSS (sav) and STATA (dta) with the respective metadata (interoperability). Publishing the data only as sav and dta might be a limitation for all researchers which don’t have access to these statistical programs. As the NEPS provides access to different statistical software packages (SPSS, R, Stata, Mplus) in the RemoteNEPS environment researchers can work there to analyze the NEPS data.”

#5. Reviewer: References:

I think the authors forgot to delete a part of the template (the introductory sentence after “References”) they used to generate the manuscript.

Response by authors:

Yes that’s right. We deleted the sentence.

Review 2

#1. Reviewer:

On page 1, the second paragraph in the Background section reads “Following the assumptions of theoretical models such as the bioecological model of development (Bronfenbrenner & Morris, 2006), emphasizing the interplay of multiple contexts in shaping children’s development, not only their own abilities and prerequisites play a role for their development but also the learning environments they grow up in and interact with”. The readability of this sentence might be enhanced if it was split into shorter sentences.

Response by authors:

Thanks for the suggestion. As suggested we splitted the sentence:

“Theoretical models such as the bioecological model of development (Bronfenbrenner & Morris, 2006) emphasize the interplay of multiple contexts in shaping children’s development. Hence not only children’s own abilities and their prerequisites play a role for their development but also the learning environments they grow up in and interact with.”

#2. Reviewer:

It could be more explicit early on in the Background section that this paper is describing the NEPS SC1 data only. I think making this clear early on will also aid readability.

Response by authors:

We added the following sentence in the first paragraph of the background section (and removed it in the later part of the section):

“To take this into account, the German National Educational Panel Study (NEPS; Blossfeld & Roßbach, 2019) includes a Newborn Cohort Study (Starting Cohort 1; SC1; e.g., NEPS Network, 2022; Weinert, Linberg, Attig, Freund & Linberg, 2016) to investigate educational trajectories and the development of competencies from early on. The research potential of these data will be introduced in the present paper.”

#3. Reviewer:

On page 3, it reads “Beginning with a child age of around 7 months and already 9 published assessment waves on the children and their families...”. Perhaps consider restructuring this sentence as “and already 9 published assessment waves...” is a bit unclear grammatically. Something like “and with 9 further assessment waves already published...” seems clearer.

Response by authors:

Thanks for the hint. The sentence mixed up some thoughts, we changed the sentence to:

“With already 9 assessment waves published on the children and their families....”

#4. Reviewer:

On page 3, in the first sentence of the Methods section, it is a bit unclear exactly what is meant by ‘individual sample’, so perhaps this could be clarified.

Response by authors:

Thanks for the carefully reading of the manuscript. Indeed, this information is not really necessary. We deleted the sentence.

#5. Reviewer:

Table 1 is very useful, but the bottom rows are very difficult to read in the current format due to the way it spills onto the second page. This might be something that gets fixed further down the line, but can the authors ensure that the table is adjusted so it fits on one page?

Response by authors:

Yes of course, we will have this in mind when we get the proof of the journal.

#6. Reviewer:

On page 6, at the end of the first paragraph, it reads “...paper-and-pencil questionnaires (PAPIs) for one parent as well as (PAPIs for educational professionals...””, but it could be clearer whether it means ‘one parent’ for each child or one parent in total across the whole cohort.

Response by authors:

We added a “of each child” to make this clear.

#7. Reviewer:

On page 6, at the beginning of the second paragraph, it mentions that “semi-standardised videotaped observational measures” were used. It would be helpful to understand a little more about what/wasn’t standardised or pointed to another source with more information.

Response by authors:

Thank for the hint. We refer now to section 2.5. where we described the measures and in this section we also link to further sources with more information into that.

#8. Reviewer:

Similarly, in the next paragraph, it mentions “design decisions” were made. Some elaboration on what the decisions were would help inform the reader.

Response by authors:

We added more information to that wave in the footnote and now also describing shortly how the sample was drawn for that subsample. We hope it’s clear now.

“To reduce the costs of the survey and the burden on the families, the second wave was planned as a CATI field. However, as early direct measures are rare in panel studies and offer high potential for research questions, a design was developed where at least about half of the panel sample was randomly selected for an additional assessment of observational measures in the households of the families. Thus, a random sample of 34 municipalities was drawn from the initial 84 municipalities (see section 2.4.). This subsample was asked to additionally participate in the household field.”

#9. Reviewer:

Also on page 6 are several references to "CAPI-field" and I think it could be clearer what the ‘field’ part is in reference to for those that might be unfamiliar with this type of design

Response by authors:

Thanks again, we made small changes to the sentence to make that more clear.

“Beginning with wave 3, all families who could not be contacted for the CAPI household survey or preferred a telephone interview instead of the personal interview in their households were converted to a CATI-survey (around 5% of the interviews took place as telephone interview across the waves).”

#10. Reviewer:

The paragraph on training interviews, again on this page, could maybe be removed as it is covered again in data quality.

Response by authors:

Yes thanks. We removed the paragraph in this section.

#11. Reviewer:

In Table 2, the title reads “Sample description (child and parent)”, but the heading in the table refers to “child” and “respondent”. If the respondent is always the parent, can the heading be renamed for consistency? If the respondent is not always the parent, does the title need to change?

Response by authors:

We changed the title. It’s one legal guardian which gives us the information in the interviews and allowed us to do the assessments with the child (and of course, there might be cases where for example the grandparents are the legal guardians). Therefore we use the term respondent.

#12. Reviewer:

On page 18, it is explained that the ELFRA-2 “isn’t included in this NEPS documentation as it is a published instrument”. Could the authors clarify what about the instrument being published prevents it from being included? Is it because it is copyrighted or is there some other issue?

Response by authors:

Yes, you have to pay for the use of the ELFRA. So we are not allowed to publish the items of the instrument in the documentation. We added the following part in the manuscript:

“As an exception, one questionnaire (the early child language checklist; ELFRA-2 (Grimm & Doil, 2006)) isn’t included in this NEPS documentation as it is a published instrument whose items are copyrighted.”

#13. Reviewer:

In general, the sheer volume of NEPS data – even just for the SC1 – is a bit overwhelming and takes a lot of time to explore. That is inevitable owing to its size, of course. One thing I found particularly helpful was the codebook on this page: https://www.neps-data.de/Portals/0/NEPS/Datenzentrum/Forschungsdaten/SC1/10-0-0/SC1_10-0-0_Codebook_en.pdf. Perhaps the authors could link to that as well as the NEPSplorer in section 3.9 with the codebook?

Response by authors:

Yes, thanks for the good idea. We added a sentence and a link to the documentation part in section 2.9. “Further, information on the content of all data sets are accessible in the codebook

of each version of the SUF (see here <https://www.neps-data.de/Data-Center/Data-and-Documentation/Start-Cohort-Newborns/Documentation> 'Codebook').”